# Research on Mechanisms of Improving Flotation Selectivity of Coal Slime by Adding Sodium Polyphosphate

Yusai Wang [1,2], Ying Zhou [1,2], Qi He [1,2], Yaowen Xing [2], Xicheng Bao [1,2], Xiahui Gui [2] and Lei Wang [2,*]

1   School of Chemical Engineering and Technology, China University of Mining and Technology, Xuzhou 221116, China
2   Chinese National Engineering Research Center of Coal Preparation and Purification, Xuzhou 221116, China
*   Correspondence: lei.wang@cumt.edu.cn; Tel.: +86-158-6398-0650

**Abstract:** A high percentage of high-ash fine slime materials can deteriorate flotation selectivity through surface covering. This covering of the surface is one of the issues that need to be addressed for efficient flotation processing of difficult-to-separate and high-ash coals. In this study, we investigated the depression effect of SPP on high-ash fine mud by flotation kinetic tests. We also revealed the mechanism of SPP depression of fine slime flotation and enhanced flotation selectivity of difficult-to-separate and high-ash coals by means of Scanning Electron Microscopy (SEM), Energy Dispersive Spectroscopy (EDS), and Atomic Force Microscopy (AFM) analyses. The results showed that under the best condition of SPP dosage, clean coal with 9.75% ash content and 76.76% yield was obtained. Compared to the blank group, the ash content of the clean coal decreased by 2.39%, while the yield was only reduced by 2.18% in the presence of SPP. The reason for this was that the addition of SPP enhanced the stripping and dispersion of the gangue from the coal particle surface. The result was a reduction in the cover of the coal grain surface and an increase in the hydrophobic sites on the coal surface, thereby depressing the non-selective flotation of the gangue and enhancing the adsorption of the collector on the coal surface. The ash content of the flotation concentrate decreased, but the yield remained almost unchanged, which was the main reason for the better performance of SPP as a depressant compared to conventional depressants.

**Keywords:** coal; flotation selectivity; slime dispersion; sodium polyphosphate; reagent adsorption





## 1. Introduction

In recent years, to maximize coal production and utilization, coal mining operations and preparations have become highly mechanized, resulting in high proportions of fine coal [1]. In the field of coal cleaning, more and more attention has been paid to the separation of fine slime. Unlike lump coal, which is cleaned based on density differences, the specific gravity differences in fine slime are low. Flotation is considered to be the most efficient method for processing fine slime. However, the reduced selectivity due to the presence of high-ash fine slime presents a challenge for flotation separation [2–4]. High-ash fine slime is mainly composed of fine-grained clay minerals and micro-minerals produced in the process of sliming. A large amount of high-ash fine slime tends to coat the surface of coal particles via electrostatic attraction and reduce surface hydrophobicity, which results in a low recovery rate of combustibles [5,6]. Moreover, due to its light weight and small particle size, the heterogeneous slime is entrained into the clean coal by non-inertial motion, further deteriorating the flotation process [7,8].

To prevent high-ash fine slime and improve flotation selectivity, some scholars have tried various depressants in the flotation process [9,10]. Sodium silicate is a depressant extensively used in coal flotation to depress quartz and silicate gangue minerals [11,12]. However, high dosage and unstable depression limit its industrial application. A residual sodium silicate in the slurry could also cause problems in the subprocesses of precipitation

and filtration of flotation concentrate and tailings [13,14]. In addition to sodium silicate, a number of natural organic polymers, including dextrin [15], starch [16], humic acid [17], guar gum [18], and cellulose [9,18] have also been used as depressants for coal flotation. Although these depressants generally have a significant depressing effect on coal while suppressing gangue, they also reduce coal floatability, leading to a large amount of combustible material in tailings. Therefore, it is urgently required to explore highly selective depressants to separate coal and gangue efficiently.

Many studies have pointed out that inorganic phosphate can disperse silicate minerals, carbonate minerals, and oxide minerals, which is believed to be because the addition of inorganic phosphates can increase the electrostatic repulsion and steric hindrance between minerals [19–22]. Theoretically, this dispersion can peel off and disperse the fine gangue that adheres to the surface of coal particles, strengthening the recovery of coal particles, and thereby reducing the ash content of clean coal and enhancing flotation selectivity. Moreover, inorganic phosphate is very easy to ionize in aqueous solutions and produces active anions to form colloidal complexions or chelate with divalent metal ions (such as $Ca^{2+}$, $Mg^{2+}$, and $Fe^{2+}$) [23]. It can reduce the surface hydrophobicity of these minerals and have a strong depressant effect, and further strengthen the selective recovery of the desired mineral. Some scholars have studied the depressing effect of inorganic phosphate in mineral processing. Wang et al. [24] found that sodium tripolyphosphate (STSP) could strongly bind to Ca sites on the surface of calcite but only weakly bind to Mg sites on the surface of magnesite. Therefore, STSP is considered an effective flotation depressant for magnesite and calcite. Kang et al. [25] studied a highly selective sodium hexametaphosphate (SHMP) system wherein SHMP was hydrolyzed into $HPO_4^{2-}$ and adsorbed on the positively charged surfaces of calcite and fluorite via electrostatic force or chelation. Based on the above literature, it is expected that sodium polyphosphate (SPP) has a good depressing effect on gangue, such as calcite and kaolinite, in high-ash hard-to-separate coals. However, the application of SPP to improve slime flotation selectivity as a highly efficient dispersant and depressant has not been extensively studied and has not been applied on a large scale.

This paper attempted to employ SPP as a depressant to efficiently float difficult-to-separate and high-ash coal. The effect of SPP on flotation selectivity was studied by flotation kinetic tests. To explore the mechanism of SPP depression, Scanning Electron Microscopy (SEM), Energy Dispersive Spectroscopy (EDS), Atomic Force Microscopy (AFM), Total Organic Carbon (TOC) measurements, and X-ray Photoelectron Spectroscopy (XPS) were used. This study provides important strategies for the utilization of difficult-to-separate and high-ash coal.

## 2. Experimental

### 2.1. Material and Reagents

The coal samples used in this study were collected from the Qiuji Coal Preparation Plant in Dezhou City, Shandong Province, China. To analyze the particle size composition of coal samples, a wet screening test was conducted with a set of standard sieves with mesh sizes of 0.500, 0.250, 0.125, 0.074, and 0.045 mm. The particle size distribution is shown in Table 1. It should be noted that the main size fraction was 0.045 mm with a 73.84% by weight, and the ash content was 30.23%. This part of the material belonged to high-ash fine slime. The results of the X-ray diffraction analysis of the gangue are shown in Figure 1. The gangue minerals were mainly quartz, kaolinite, and calcite. Kaolinite is a clay mineral that is prone to sliming [26].

**Table 1.** Particle size distributions of the coal samples.

| Size (mm) | Weight (%) | Ash (%) | Cumulative Oversize | | Cumulative Undersize | |
|---|---|---|---|---|---|---|
| | | | Weight (%) | Ash (%) | Weight (%) | Ash (%) |
| −0.500 + 0.250 | 0.37 | 6.57 | 0.37 | 6.57 | 100.00 | 25.30 |
| −0.250 + 0.125 | 2.56 | 6.73 | 2.93 | 6.71 | 99.63 | 25.37 |
| −0.125 + 0.074 | 11.74 | 8.54 | 14.67 | 8.17 | 97.07 | 25.86 |
| −0.074 + 0.045 | 11.49 | 15.51 | 26.16 | 11.40 | 85.33 | 28.25 |
| −0.045 | 73.84 | 30.23 | 100.00 | 25.30 | 73.84 | 30.23 |
| Total | 100.00 | 25.30 | | | | |

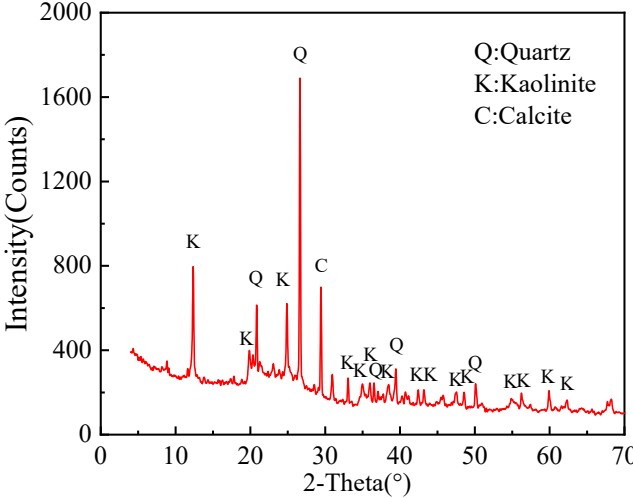

**Figure 1.** X-ray diffraction analysis of gangue.

Diesel oil was purchased from Sinopec and used as a collector. Secondary octanol was utilized as the frother. SPP was purchased from Shanghai McLean Biochemical Co., Ltd. (Shanghai, China) and was used as a depressant. Considering its solubility and the interaction efficiency with gangue particles during flotation, the SPP was configured as a 5% aqueous solution and added to the slurry before the collector.

*2.2. Methods*

2.2.1. Flotation Kinetics Experiments

Flotation kinetics tests were carried out in an XFD-1.0 $dm^3$ laboratory flotation machine. The impeller rotation was kept constant at 1800 rpm. The solid content and airflow rate were kept constant at 80 $g/dm^3$ and 0.1 $m^3/h$, respectively. The coal was prewetted for 2 min. After the prewetting process, SPP, collector, and frother were added to the pulp step-by-step. The dosage of diesel oil and secondary octanol were kept constant at 400 g/t and 100 g/t, respectively. The dosages of SPP were 0, 1000, 2000, 3000, and 4000 g/t. The flotation concentrate was collected at periods of 0–30, 30–60, 60–120, 120–180, and 180–300 s. After filtration and drying (80 °C), each product was weighed to calculate the flotation yield. For the ash analysis, the drying samples were first mixed evenly, then, approximately 1 g of the sample was burned in an oven at 815 °C for 2 h.

The effect of SPP on flotation selectivity was evaluated using the Fuerstenau upgrading curve, which was plotted according to the results of the flotation kinetics experiments. The selectivity of the flotation can be characterized by the flotation selectivity index (*k*). The kinetic equations relate the recoveries to the time of the two components in separation products, and when combined eliminate the time parameter [27,28]. Commonly, the flotation kinetics of both the combustible and ash materials can be successfully described by a classical first-order equation, as shown in Equations (1) and (2) [29]. Equations (1) and (2) yield Equation (3), which is a mathematical formula for the approximation of the Fuerstenau

curves. When the values of $\varepsilon_\infty$ and $\varepsilon_{a\infty}$ were approximate to 100%, and $k$ (Fuerstenau selectivity index) was chosen to represent $k_1/k_2$, Equation (3) was then transformed to Equation (4).

$$\varepsilon = \varepsilon_\infty \left(1 - e^{-k_1 t}\right) \tag{1}$$

$$\varepsilon_a = \varepsilon_{a\infty} \left(1 - e^{-k_2 t}\right) \tag{2}$$

$$\varepsilon = \varepsilon_\infty \left[1 - \left(\frac{\varepsilon_{a\infty} - \varepsilon_a}{\varepsilon_{a\infty}}\right)^{k_1/k_2}\right] \tag{3}$$

$$\varepsilon = 100 \left[1 - \left(\frac{100 - \varepsilon_a}{100}\right)^{k}\right] \tag{4}$$

where $\varepsilon$ is the recovery rate of combustible matter and and $\varepsilon_a$ is the ash in the concentrate. $\varepsilon_\infty$ and $\varepsilon_{a\infty}$ represent the maximum value. $k_1$ and $k_2$ are the first-order flotation rate constants of combustible and ash-forming material, and $t$ is the flotation time.

### 2.2.2. Particle Size Measurements

A laser particle sizer (BT-9300S, Dandong Baite Instrument Co., Ltd., Shenyang, China) was used to characterize the apparent particle size of the coal samples. Laser particle size analysis can reflect the apparent particle size of slime particles [30,31]. The effect of SPP on the slime coating and dispersion of high-ash fine slime was studied according to the change of particle size abundance before and after the action of SPP. One gram of coal sample was weighed and configured with deionized water as a suspension (mass fraction 2%), as, according to the results of the flotation kinetics test, the optimum dosage of SPP was determined as 1000 g/t. Controlled tests were carried out by controlling the presence of SPP in the slime suspension. The suspension was sonicated for 3 min and then subjected to laser particle size testing, and the measurements were averaged several times to ensure the accuracy of the test.

### 2.2.3. SEM-EDS Analysis

An SEM system (EM-30, COXEM Co., Ltd., Yuseong-gu, Daejeon, South Korea) equipped with an energy dispersive X-ray spectroscopy (EDS) attachment was used to qualitatively analyze the coarse-grained coal and the fine-grained components attached to its surface and understand the occurrence state of high-ash fine slime. The preparation method of the SEM test sample involved adding SPP to the coal slurry with a concentration of 80 g/L such that the dosage of SPP is 1000 g/t. After 3 min of action, an appropriate amount of mineral slurry was evenly dripped onto a glass slide and dried in a dust-free environment.

Firstly, SEM scanning was carried out at the magnification of 1000 times to understand the apparent morphology of the particles. To accurately obtained more abundant microscopic information on the sample surface, representative particles were selected for point scanning in the scanning area at 2000 times magnification. EDS point scanning was carried out twice on the surface of the coarse-grained coal and the corresponding energy spectrum diagrams were denoted as A and B. The spectrum diagrams C, D, E, and F were the energy spectrum diagrams of fine particles on the surface of the coarse-grained coal. The contents of the main elements C, O, Al, Si, and Ca in many kinds of particles were quantitatively analyzed according to the energy spectrum. The high-ash fine slime and fine coal were distinguished according to the content of the main elements, which directly represented the coverage level of the high-ash fine slime. At the same time, the influence of the depressant on the slime coating was investigated.

### 2.2.4. AFM Interaction Force Measurements

To study the effect of SPP on the interaction between gangue particles, glass beads, and glass substrate was selected to represent gangue particles. The interaction force

measurements between solids were tested in pure water and 1% SPP solution, respectively. Under monitoring with an optical microscope, a colloidal probe was prepared, and a glass bead with a diameter of approximately 40 μm was fixed at the end of the needle-free probe with epoxy resin. To reduce the test error, the same substrate and colloidal probe are selected throughout the test. The colloidal probe injection and withdrawal rate was 400 nm/s. The colloid probe was mounted on a special test liquid pool (O-ring) and the O-ring was mounted to form a closed liquid cell. Deionized water or SPP solution was sucked slowly into the cell using a syringe in case of the introduction of the air bubble. When the experiments in deionized water were finished, the SPP solution was sucked indirectly to replace it. A complete operation cycle of force measurements by the colloid probe–surface approach and separation was performed [32].

### 2.2.5. TOC Measurements

In order to better judge the effect of SPP on the adsorption of the collector on the surface of the slime, the difference in TOC content between the collector and coal sample before and after adsorption was employed in this study to calculate the adsorption amount of the collectors. The concentration of residual collector in the solution was characterized by measuring the content of TOC in the solution. The collector was mixed with water as the initial solution. The interaction between collector and coal can be divided into two cases: acting alone with coal and working together with SPP. The collector will remain in the slurry after being adsorbed and saturated with coal. The difference between the initial concentration and the residual agent concentration was the content of the adsorbed collector of the coal sample [33]. The greater the difference, the more adsorbed collectors on the coal surface. To maintain the consistency of the flotation experiment, the slurry concentration remained at 80 g/L. The dosage of the collector was 400 g/t and the amount of SPP was 1000 g/t. The mixing process was carried out on a magnetic stirrer and the mixture was stirred for 10 min to allow the coal sample and collector to fully adsorb. After the coal slurry was stirred, it was centrifuged. The centrifugal liquid was filtered by a PES water-based filter (pore size of 0.8 μm) to obtain the filtrate as the remaining collector content in the solution. The initial solution and the filtrate were injected into the TOC analyzer. The organic carbon content was measured at 700 °C by combustion catalytic oxidation. To ensure the accuracy of the test, three parallel tests were carried out and the results of the TOC content were averaged thrice.

### 2.2.6. Contact Angle Measurements

The contact angle was used to reflect the hydrophobicity difference between the coal surface and the adsorption of collector on the coal surface. The coal samples treated with collector alone and treated with depressant and collector together were pressed into smooth flakes by a tablet press. The static contact angle of the coal sample was measured using the dropping method. The whole process, from when the droplets touched the coal surface to stable adsorption, was recorded by a high-speed dynamic camera.

### 2.2.7. XPS Analysis

The type of mineral surface element species under different reagent conditions was measured by XPS (ESCALAB 250Xi, Thermo Scientific, Waltham, MA, USA). Casa XPS software was utilized to fit and analyze the XPS peaks. Binding energy calibration was performed by setting the C1s hydrocarbon peak at 284.6 eV [34].

## 3. Results and Discussion

### 3.1. Flotation Test Results

Figure 2 shows the flotation results under various SPP dosages. The larger the k value, the better the selectivity. By comparing the k values, the flotation selectivity increased and then decreased with the increase in SPP dosage. When the SPP was 1000 g/t, the flotation selectivity was the best and the k value reached its maximum value of 5.20.

The correlation coefficient $R^2 = 0.9776$, indicating that the result was fitted well by the Fuerstenau upgrading curve. When the dosage of SPP was 1000 g/t, the ash content of clean coal decreased from 12.14% to 9.75%, while the recovery rate of combustibles only reduced from 91.99% to 91.19%. This indicated that adding SPP had no significant effect on the recovery of combustibles but showed an excellent depression of fine gangue slime to improve the flotation selectivity in difficult-to-separate and high-ash coal flotation.

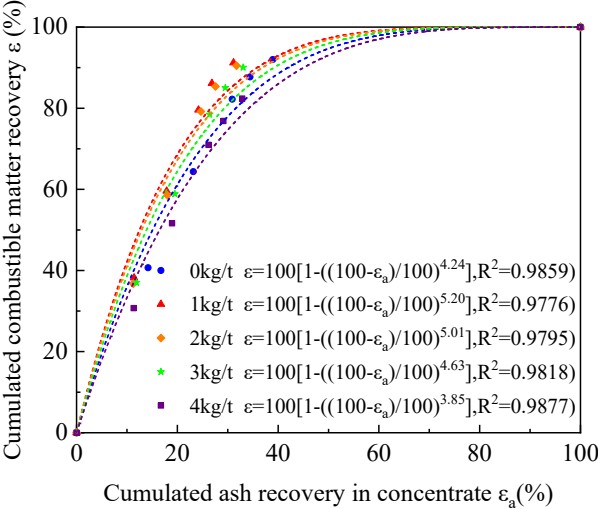

**Figure 2.** The Fuerstenau upgrading curve of difficult-to-separate and high-ash coal flotation under the action of SPP.

### 3.2. Dispersion Behavior of High-Ash Fine Slime

3.2.1. Laser Particle Size Analysis

Figure 3 shows the change in the apparent particle size of the slime in the presence and absence of SPP. The overall particle size of coal was small, and most particles were less than 74 μm. The particle size distribution was mainly in the range of 5~45 μm. Comparatively, adding SPP increased the proportion of fine-grained particles. The changes in apparent particle size were characterized by D10, D50, and D90 under different depressant conditions, and the results are shown in Table 2. The smallest particle was approximately 1 μm. Without SPP depressant, the D50 and D90 of the coal sample were 15.44 and 57.88 μm, respectively. After being treated with SPP, the coal sample had a decreased D50 (11.25 μm) and D90 (41.17 μm). This was probably due to the preferential adsorption of SPP on the surface of gangue particles and a reduced slime coating on the coal particles [23].

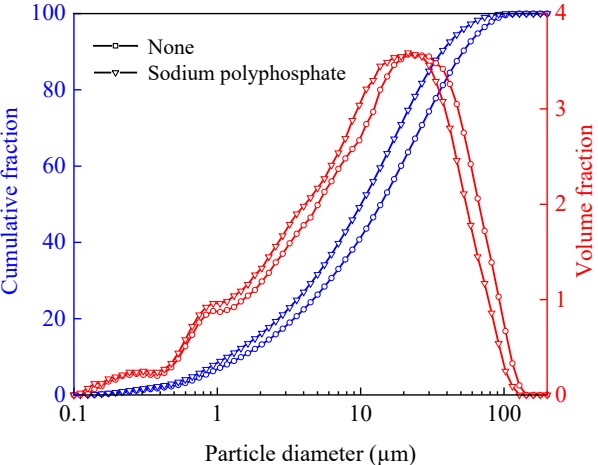

**Figure 3.** Particle size distribution curves of coal slime under the action of SPP (SPP:1000 g/t).

**Table 2.** The grain size parameters of samples.

| Condition | D10 (μm) | D50 (μm) | D90 (μm) |
|---|---|---|---|
| None | 1.65 | 15.44 | 57.88 |
| With SPP | 1.32 | 11.25 | 41.17 |

### 3.2.2. SEM-EDS Analysis Results

Figure 4 shows the SEM images of the coal samples before and after adding SPP. The content of fine particles in the raw coal was high, and most particles were below 50 μm. Fine particles mainly existed in the form of floc groups or on the surface of coarse particles. In the presence of SPP, the surface of the coarse particles was smoother, and the number of attached fine particles was greatly reduced. Fine particles are mainly well dispersed in the pulp phase.

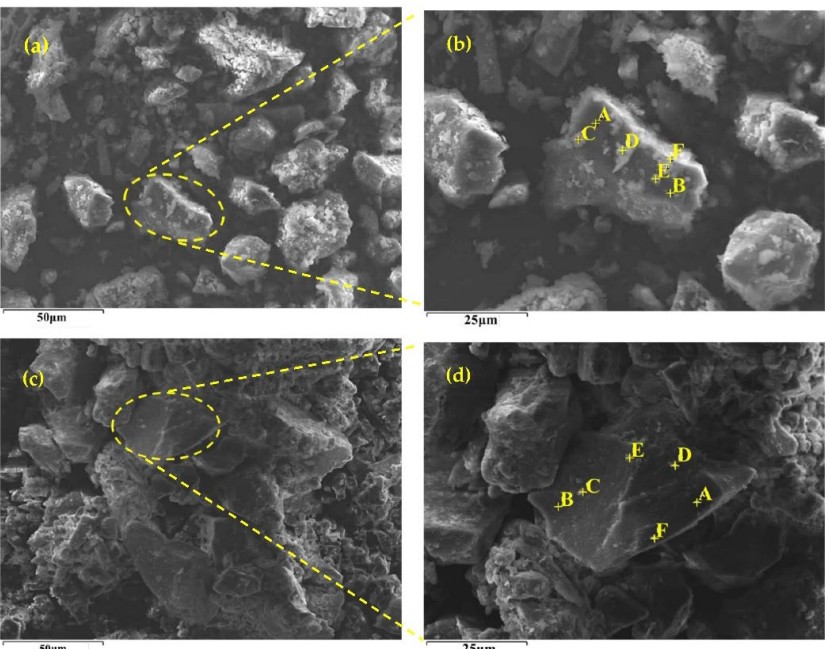

**Figure 4.** SEM images of raw coal magnified 1000 times (**a**); raw coal magnified 2000 times (**b**); coal samples were treated with SPP at 1000 times magnification (**c**); and coal samples were treated with SPP at 2000 times magnification (**d**). Letters A-F are the EDS point scanning positions respectively.

To qualitatively analyze the covering of fine particles on the coal surface before and after the action of SPP, EDS spectra were measured, and the results are shown in Table 3. The carbon elements in the spectra of raw coal A and B were as high as 82.71% and 82.28%, respectively, indicating that the main component of this coarse particle was coal. The contents of Al and Si elements were high in the spectra of most fine particles (except for spectrum E). According to the XRD analysis results of the raw coal, the main sources of Al and Si elements were gangue minerals, such as kaolinite and quartz. Therefore, the surface coating on coal slime was mainly fine gangue particles. After adding SPP, the contents of Al and Si were decreased (except for spectrum D). The contents of carbon element were similar to coarse-grained coal, suggesting that coal particles were attached to the surface of coarse-grained coal with the action of SPP.

In other words, the covering of high-ash fine gangue was significantly reduced on the surface of coal. SPP could change the electrical properties of the surface of fine mineral particles so that the fine mineral particles originally attached to the coarse particles were dispersed in the suspension system. This is consistent with the results of the laser particle size analysis.

**Table 3.** EDS point scanning spectral analysis of coal sample.

| Sample | Name of Spectra | Element | | | | |
|---|---|---|---|---|---|---|
| | | C (%) | O (%) | Al (%) | Si (%) | Ca (%) |
| Raw coal | A | 82.71 | 13.14 | 0.59 | 0.44 | 0.19 |
| | B | 82.28 | 16.19 | 0 | 0 | 0.17 |
| | C | 52.36 | 32.82 | 5.22 | 7.44 | 0.14 |
| | D | 65.27 | 27.31 | 2.42 | 2.92 | 0 |
| | E | 73.44 | 21.47 | 1.19 | 1.61 | 0.08 |
| | F | 43.06 | 40.86 | 5.67 | 6.65 | 0.66 |
| Coal under the action of SPP | A | 83.69 | 12.45 | 0.47 | 0.12 | 0.08 |
| | B | 80.15 | 14.76 | 0.90 | 0.63 | 0.14 |
| | C | 81.38 | 13.86 | 0.72 | 0.61 | 0.23 |
| | D | 60.00 | 16.37 | 2.79 | 4.85 | 2.00 |
| | E | 84.02 | 11.75 | 0.65 | 0.44 | 0.11 |
| | F | 79.40 | 9.83 | 1.20 | 0.46 | 0.42 |

### 3.2.3. Interaction Force between Hydrophilic Particles with SPP

The interaction forces between hydrophilic particles in different solutions are shown in Figure 5. At large separation distances, no interaction force was found between the hydrophilic particles. When the particles approached each other, a strong repulsive force was observed [35] with no obvious adhesion force. The repulsive force increased with decreasing separation distance. Different particle distances were required for repulsion to occur in SPP solution and deionized water. A repulsive force was observed at a 4-nm separation distance between glass beads and glass substrates in deionized water. In the SPP solution, the action range of repulsive force between hydrophilic particles became larger (6 nm). At the same time, the repulsive force in the SPP solution was always more significant than that in deionized water. This suggests that SPP could increase the repulsion between hydrophilic particles by changing the electrical properties of the surface. This could be one of the main reasons for SPP to improve flotation selectivity. The gangue particles in coal slime were mainly silicate minerals. In this study, the spherical and flat glass were used instead of gangue particles, which may not consider the effect of shape and surface roughness. However, previous studies have shown that glass had good consistency with real minerals [36].

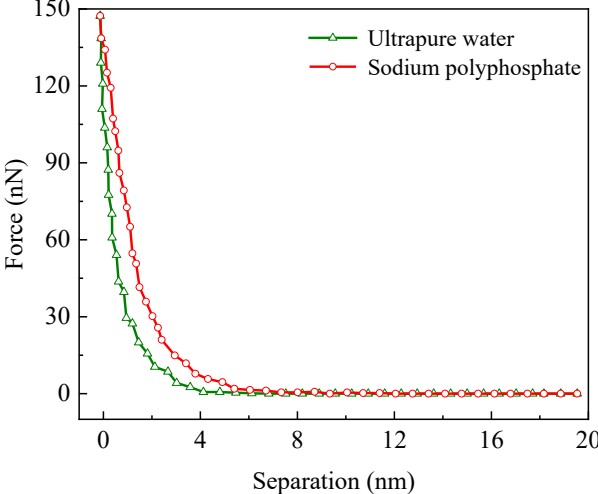

**Figure 5.** Interaction force test between hydrophilic particles in different solution environments.

Through the size analysis in SEM-EDS and interaction force measurement, it was clear that adding SPP increased the repulsion between gangue particles and enhanced the

dispersion of high-ash fine slime in coal slurry. In addition, SPP reduced the coverage of fine slime on the coal surface, which also contributed to the flotation selectivity improvement.

### 3.3. Effect of Depressants on the Adsorption of Collector

#### 3.3.1. Adsorption Capacity of the Collector

The adsorption characteristics of the collector on the slime surface were measured by TOC tests, and the results are shown in Figure 6. The dosage of the collector was 400 g/t and the amount of SPP was 1000 g/t. In the absence of SPP, the adsorption capacity of the collector on the slime surface was 0.3153 mg/g. However, under the combined action of SPP and collector, the adsorption capacity of the collector was increased to 0.3169 mg/g. Our result indicated that SPP could promote the adsorption of collectors on the coal particle surface, unlike conventional depressants, which tend to reduce adsorption activity.

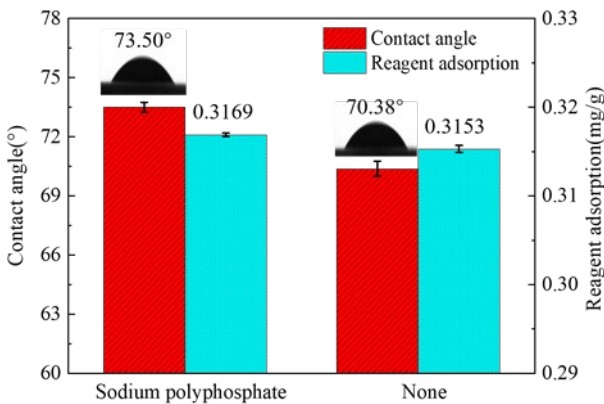

**Figure 6.** Collector adsorption and contact angle measurement results.

The adsorption of reagents generally alters the hydrophobicity of the coal particle surface. The contact angles of coal particles treated with SPP were also measured to verify the effect of SPP on the hydrophobicity. As shown in Figure 6, the contact angle was 70.38° when the raw coal was treated with 400 g/t collector. Under the combined action of 1000 g/t SPP and 400 g/t collector, the contact angle increased by 3.12° to 73.50°. SPP promoted the adsorption of the collector in coal slime and enhanced the hydrophobicity of coal slime, consistent with the results of the flotation experiment.

#### 3.3.2. XPS Analysis Results

Figures 7 and 8 show the XPS survey spectra of coal slime treated with diesel oil without and with SPP, respectively. The peak positions of C, O, Si, and Al elements did not change before and after adding SPP. However, the concentrations of C increased from 47.46% to 50.09%, while that of O decreased from 52.54% to 49.91%. C1s peak fitting was performed to further analyze the chemical groups on the coal surface, and the results are shown in Table 4. The binding energies at 284.60, 285.30, and 286.30 eV corresponded to C-C/C-H, C-O, and C=O, respectively. When the collector acted alone with the coal sample, the contents of C-C/C-H, C-O, and C=O were 66.93%, 25.79%, and 7.28%, respectively. However, when SPP and collector acted together, the contents of the C-C/C-H groups increased by 3.14%, while those of the C-O and C=O groups decreased by 2.43% and 0.71%. The increment in C concentration was because the surface of the coal particles was covered with more reagents, indicating that SPP promoted the adsorption of the collector on the coal surface. The decrease in the content of oxygen-containing functional groups revealed that the hydrophobicity of coal was further enhanced.

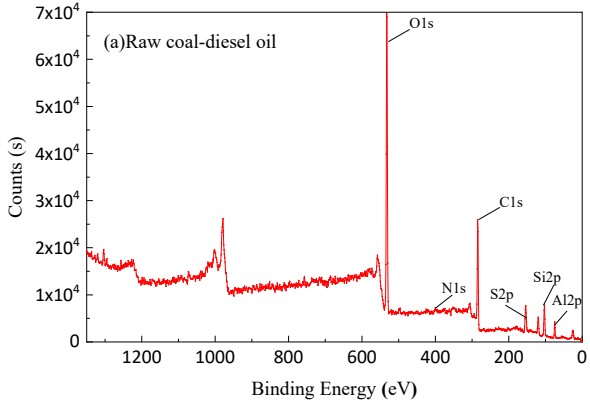
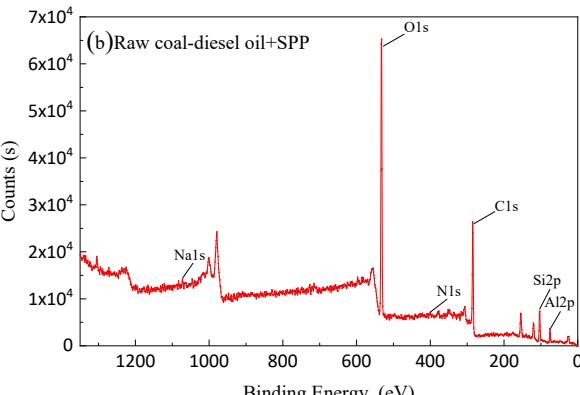

**Figure 7.** XPS wide spectrum of coal samples with different conditions: (**a**) coal was treated with diesel oil; (**b**) coal was treated with diesel oil and SPP.

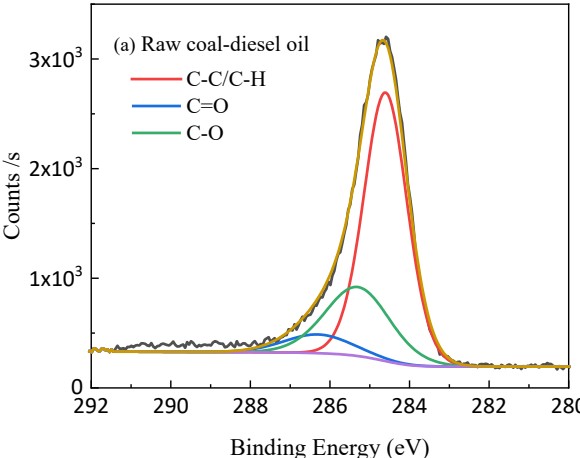
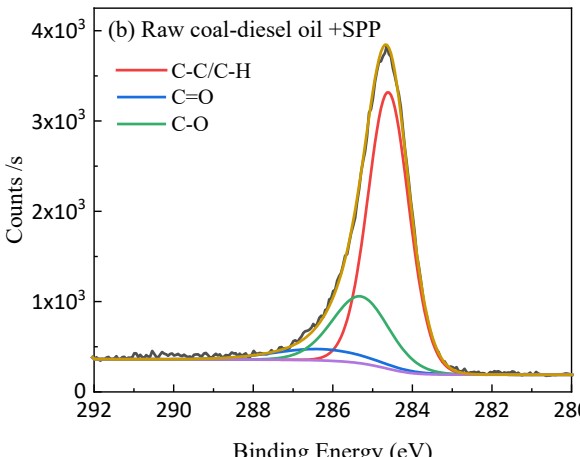

**Figure 8.** Fitting of the C1s peaks of coal samples with different conditions: (**a**) coal was treated with diesel oil; (**b**) coal was treated with diesel oil and SPP.

**Table 4.** Relative elemental atomic concentrations of coal samples with different samples.

| Samples | C-C/C-H (%) | C-O (%) | C=O (%) |
|---|---|---|---|
| Raw coal-diesel oil | 66.93 | 25.79 | 7.28 |
| Raw coal-diesel oil + SPP | 70.07 | 23.36 | 6.57 |

Figure 9 shows the mechanism for SPP promoting the separation between the slime and the coal. SPP can interact with the surface of slime to improve its dispersion. SPP is preferentially adsorbed on the surficial active sites of gangue particles, reducing the coating of gangue on coal, which is beneficial to the adsorption of collectors on the coal surface. The reduction in the cover of the coal grain surface and an increase in the hydrophobic sites on the coal surface also contributed to an improved flotation performance. Therefore, flotation selectivity is improved mainly because SPP enhances particle dispersion, prompting the adsorption of the collector and depressing the slime coating on the coal.

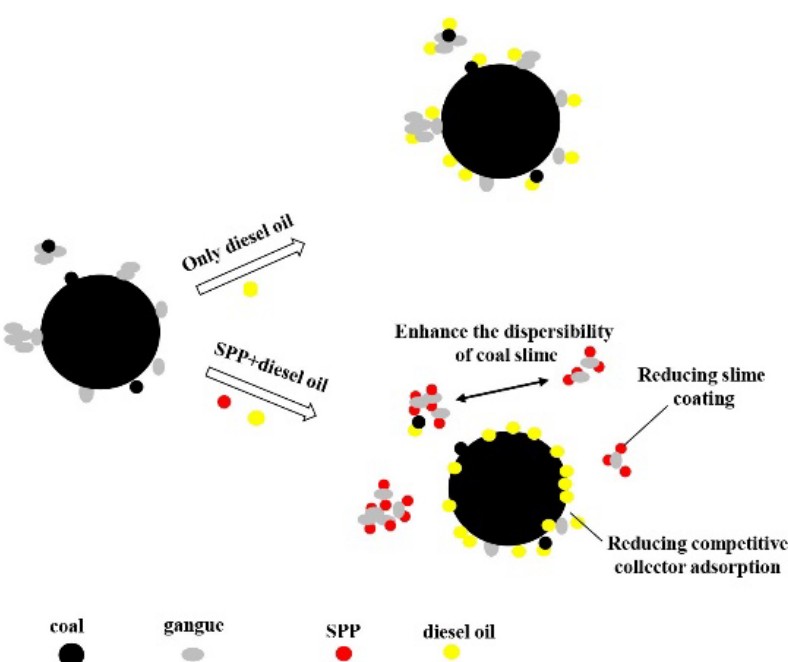

**Figure 9.** Schematic diagram of dispersion mechanism of fine slime strengthened by SPP.

**4. Conclusions**

The main conclusions that can be drawn from this work are as follows:

(1) SPP had a negligible effect on combustible recovery, but it could significantly reduce the ash content of clean coal and improve the quality of clean coal. Compared with the test results of the blank group without SPP depressant, adding 1000 g/t SPP could decrease the ash content of clean coal by 2.39%, while combustible recovery decreased by only 0.8%.

(2) SPP changed the electrical properties of gangue particles and facilitated their dispersion in pulp under a strong repulsive force. The coverage of high-ash fine slime on the surface of coarse coal was significantly reduced.

(3) Active sites on the surface of gangue particles were preferentially adsorbed by SPP, which decreased the possibility of gangue particles being absorbed on coal. The reduction in the cover of the coal grain surface and an increase in the hydrophobic sites on the coal surface also contributed to an improved flotation performance.

**Author Contributions:** Conceptualization, Y.W. and Y.Z.; methodology, Y.W.; validation, Y.W., Y.X. and Q.H.; investigation, Y.W.; resources, L.W.; data curation, Y.W. and X.B.; writing—original draft preparation, Y.W.; writing—review and editing, Y.W. and L.W.; visualization, Y.W. and Q.H.; supervision, Y.W. and X.G.; funding acquisition, Y.W. and L.W. All authors have read and agreed to the published version of the manuscript.

**Funding:** This research was funded by National Natural Science Foundation of China No. U1903207, Zhangjiagang City Industry-University-Research Institute Pre-research Project No. ZKCXY2142.

**Data Availability Statement:** Not applicable.

**Acknowledgments:** We thank Erfa Ding (China University of Mining and Technology), Qinshan Liu (China University of Mining and Technology) and Jincheng Liu (China University of Mining and Technology) for their assistance in interfacial chemical analysis. We also thank the associate editor and the reviewers for their helpful recommendations and constructive suggestions.

**Conflicts of Interest:** The authors declare no conflict of interest.

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
