# Peer review of "Research on Mechanisms of Improving Flotation Selectivity of Coal Slime by Adding Sodium Polyphosphate"

_minerals, doi:10.3390/min12111392_

Round 1
Reviewer 1 Report
In this manuscript (minerals- 1948221-peer-review-v1), the authors investigated “Depression properties of fine slime materials in coal by sodium polyphosphate for the selective flotation of coal”. The study is in the scope of journal (Minerals), but the language of the manuscript is poor. The following points should be considered before reviewing and publication.
1. In the present form, it is difficult to focus on the scientific merits of the study. See the following sentence (line 97, section 2.2.1)
“The pulp was prewetted for 2 min.”: But, in fact, pulp (a mixture of solid and water here) cannot be wetted or prewetted.
A major re-write with the assistance of a native English speaker or professional institution is essential. I ended up correcting and marking some words and sentences so that the authors should benefit from these corrections and make corrections/revisions accordingly.
2. The manuscript is weak in scientific terminology (see the following examples)
a) In the Table 1: Not “positive cumulative” but “cumulative oversize”, not “negative cumulative” but “cumulative undersize” or not “Yield(%)” but “Weight (%)”.
b) Title: Not “Inhibition” but “Depression”
3. Additional mistakes or contradictions in the manuscript were highlighted in the reviewed manuscript (the attached pdf file).

Author Response
Reviewer #1: In this manuscript (minerals- 1948221-peer-review-v1), the authors investigated “Depression properties of fine slime materials in coal by sodium polyphosphate for the selective flotation of coal”. The study is in the scope of journal (Minerals), but the language of the manuscript is poor. The following points should be considered before reviewing and publication.
- In the present form, it is difficult to focus on the scientific merits of the study. See the following sentence (line 97, section 2.2.1)“The pulp was pre-wetted for 2 min.”: But, in fact, pulp (a mixture of solid and water here) cannot be wetted or pre-wetted.
Response: “The pulp was pre-wetted for 2 min” has been revised as “The coal was pre-wetted for 2 min”.
A major re-write with the assistance of a native English speaker or professional institution is essential. I ended up correcting and marking some words and sentences so that the authors should benefit from these corrections and make corrections/revisions accordingly.
Response: We thank the reviewer for the English expression improvement. We have made changes to the expressions in whole manuscript. as pointed out by the reviewer.
- The manuscript is weak in scientific terminology (see the following examples)
- a) In the Table 1: Not “positive cumulative” but “cumulative oversize”, not “negative cumulative” but “cumulative undersize” or not “Yield (%)” but “Weight (%)”.
- b) Title: Not “Inhibition” but “Depression”
Response: a) We revised the scientific terminology in the manuscript. In the revised manuscript, “Cumulative Oversize”, “Cumulative Undersize” and “Weight” are used. It is worth mentioning that in the literature, “yield” is also used to characterize the ratio of a product weight to the total sample weight [1,2].
- b) We revised the title for the manuscript. The title has been changed to “Research on Mechanisms of Improving Flotation Selectivity of Coal Slime by Adding Sodium Polyphosphate”. The reason for this change is that we agreed that the original title “Selective Inhibition of Fine Slime in Coal Flotation with Sodium Polyphosphate” would mislead readers. It is known to all that in coal flotation depressants would also reduce the combustible matter recovery to some extent. This study, however, indicates that with sodium polyphosphate only the recovery of ash materials was depressed, and the recovery of combustible materials was not influenced. This “selective” word is used for the ash materials and combustible matter, and that is why we call it “selective inhibition (depression). But there is possibility that other readers would take the “selective” as that the depression effect of sodium polyphosphate on a certain gangue mineral or some gangue minerals. To make it more precise, we changed the title of this manuscript.
- Additional mistakes or contradictions in the manuscript were highlighted in the reviewed manuscript (the attached pdf file).
Response: Thanks for the valuable comments. These are all addressed in the revised manuscript.

Reviewer 2 Report
The paper "Selective Inhibition of Fine Slime in Coal Flotation with Sodium Polyphosphate" is rich in experimental and analytical results. The manuscript was properly prepared with numerous references to literature. However, the amount of SPP depressant used in the amounts of 0, 1000, 2000, 3000 and 4000 g/t is puzzling? The more that authors mention that the best results were obtained for dose of 1000 g/t. Why has the performance of system not been investigated at doses lower than 1000 g/t? From an economic point of view, and possibly from the obtained results, it would be justified.
Other comments:
• Line 83 - there is: Kaolin was a clay mineral (...); should be: Kaolinite ...
• Figure 2, 3, 5 - data markers should be slightly larger;
• Line 286 - there is "As shown in Fig.9 (...)"; should be: "As shown in Fig.6 (...);
• Section 3.3.1 - what was the concentration of collector and SPP when the adsorption and contact angle was tested?
• Figure 8 - Y axis description is missing;
• Figure 9 - unreadable descriptions, the drawing could be enlarged;
• Line 406 - there is: ZHANG… GONG…; should be: Zhang… ..Gong… .
Author Response
Reviewer #2: The paper "Selective Inhibition of Fine Slime in Coal Flotation with Sodium Polyphosphate" is rich in experimental and analytical results. The manuscript was properly prepared with numerous references to literature. However, the amount of SPP depressant used in the amounts of 0, 1000, 2000, 3000 and 4000 g/t is puzzling? The more that authors mention that the best results were obtained for dose of 1000 g/t. Why has the performance of system not been investigated at doses lower than 1000 g/t? From an economic point of view, and possibly from the obtained results, it would be justified.
Response: In this study, we have not explored the depressing effect of SPP below 1000 g/t dosage. The selection of “0, 1000, 2000, 3000 and 4000 g/t” of SPP in the study was because that the flotation selectivity improvement is not obvious when the amount of inorganic phosphate is less than 1000 g/t, as indicated by other studies [3-5]. In this paper, we focus on whether SPP is effective for depressing ash materials in coal flotation and if so how SPP promotes the flotation selection. In the future, the depressing effect of SPP below 1000 g/t dosage will be further examined for the reagent saving purpose.
- Line 83 - there is: Kaolin was a clay mineral (...); should be: Kaolinite ...
Response: Kaolinite has been used in the revised manuscript.
- Figure 2, 3, 5 - data markers should be slightly larger;
Response: Data markers in Figures 2, 3, and 5 has been revised based on the comment.
- Line 286 - there is "As shown in Fig.9 (...)"; should be: "As shown in Fig.6 (...);
Response: This typo has been corrected in the revised manuscript.
- Section 3.3.1 - what was the concentration of collector and SPP when the adsorption and contact angle was tested?
Response: The information has been added to the revised manuscript:
“The dosage of the collector was 400 g/t and the amount of SPP was 1000 g/t.”
“As shown in Fig.6, the contact angle was 70.38° when the raw coal was treated with 400g/t collector. Under the combined action of 1000 g/t SPP and 400 g/t collector, the contact angle increased by 3.12° to 73.50°.”
- Figure 8 - Y axis description is missing;
Response: We have revised figure 8 in the revised manuscript:
- Figure 9 - unreadable descriptions, the drawing could be enlarged;
Response: Figure 9 has been enlarged in the revised manuscript to make the descriptions readable.
- Line 406 - there is: ZHANG… GONG…; should be: Zhang… ..Gong… .
Response: These have been corrected in the revised manuscript.

Reviewer 3 Report
The article “Selective Inhibition of Fine Slime in Coal Flotation with Sodium Polyphosphate” was revised. The manuscript includes experimental work about the role of sodium polyphosphate in the coal flotation system. There is a good presentation of results; however, some aspects must be improved. My comments for the authors are the following:
1. Introduction section. This section was focused on the flotation process of coal slime. Which other methods can be used to clean coal slime? What are the flotation process's advantages and drawbacks in contrast with the other processes?
2. Lines 48-52. The authors have indicated that SPP is selective to divalent metal ions such as Ca2+. After this, in lines 63-64, the authors claimed that “…it is expected that SPP has a good inhibiting effect on gange, such as calcite and kaolinite”. According to Figure 1, X-ray diffraction analysis, the gange contains quartz, kaolinite, and calcite. The SPP role as a depressant of calcite can be deduced; how acts the SPP over the kaolinite? Can it act selectively over the calcite?
3. Table 1. It seems the columns of negative cumulative are unnecessary because the positive cumulative data is already included.
4. Figure 1. Some peaks in the diffraction pattern do not have an indexation, for instance: those between 30 and 40° or those for 2theta >50°. Are there other phases in gangue?
5. Section 2.2.1 Flotation kinetics experiments. How many times was the experiment performed?
6. Line 113. What is means ??∞? This is not included.
7. The same units must be used in manuscript. For example, 1000 g/t (in line 100) and 1 kg/t (in line 128). The SPP content was expressed in g/t or mg/L. This is difficult for the analysis of results. Figure caption 1 includes that SPP used is 80 mg/L, which is the equivalent value in g/t or kg/t?
8. Figure 1 must be located after being mentioned in the text.
9. Figure 2. Why done a model of the experimental data? Is this predictive? This is not discussed. It is notable in the graph that model lines show a deviation of experimental data even though the R2 is around 0.98; which is the error in the model?
10. Lines 201-202. “…By comparing the k values, the flotation selectivity increased and then decreased with the increase of SPP dosage”. In this case, how can be explained this behavior of the SPP on selectivity? For high SPP content (4000 g/t), the k value is lower even without SPP. How acts the SPP in gangue?
11. Figure 3. SPP 80 mg/L at what amount of depressant corresponds? 1000g/t?
12. Some figures have low quality and must be improved for better visualization. For example Figure 4, the letters of the EDS points are not visible like the annotation of dimension. It is recommended that the contrast in the text be improved. Figure 9 it is not distinguished the text; the schema must also be improved.
13. Table 3 and text in lines 238-239. The authors describe that Al and Si content is diminished (except in spectrum D). These elements are associated with kaolinite and quartz; how is explained the role of SPP over these compounds? In all EDS results, the same elements are identified; what about the selectivity of SPP by calcite?
14. Figure 4. The SEM images (c) and (d) correspond to samples after modification using SPP; this demonstrates the modification of surficial characteristics; nevertheless, How do the particles changes after the flotation process?
15. Figure 5. Which is the amount of the SPP used for this experimentation?
16. In section 3.3.1 Adsorption capacity of the collector. The text describes that 25.22 mg/L and 25.35 mg/L of collector are adsorbed in slime surface without and with the SPP use. The adsorption capacity must be reported in reference to solid mass (mg/g), not in concentration (solution).
17. Figure 6. For this graph, which is the content of SPP used? Is the contact angle affected by the amount of SPP used?
18. Line 286. Figure 9 seems incorrect.
19. Which are the contents of calcite, kaolinite, and quartz in gangue before and after the flotation process using the SPP? A characterization of concentrate and tailings after the flotation using SPP must be included to establish if the use of depressant is significant, as is claimed in the abstract and conclusions.
20. What is the maximum content of gangue desirable in coal? Does an improvement in 2.39% of impurities justify the flotation process?
Author Response
Reviewer #3: The article “Selective Inhibition of Fine Slime in Coal Flotation with Sodium Polyphosphate” was revised. The manuscript includes experimental work about the role of sodium polyphosphate in the coal flotation system. There is a good presentation of results; however, some aspects must be improved. My comments for the authors are the following:
- Introduction section. This section was focused on the flotation process of coal slime. Which other methods can be used to clean coal slime? What are the flotation process's advantages and drawbacks in contrast with the other processes?
Response: Based on the comment, we have introduced the methods for coal slime processing. The revised contents are as follows:
In recent years, to maximize coal production and utilization, coal mining operations and preparations have been highly mechanized, resulting in high proportions of fine coal [6]. In the field of coal cleaning, more and more attention has been paid to the separation of fine slime. Unlike lump coal which is clean based on density difference, the specific gravity difference of fine slime is weakened. Flotation is considered to be the most efficient method for processing fine slime. However, the reduced selectivity due to the presence of high-ash fine slime presents a challenge for flotation separation. The high ash fine slime is mainly composed of fine-grained clay minerals and micro-minerals produced in the process of sliming. Large amount of high ash fine slime tends to coat the surface of coal particles via electrostatic attraction and reduce the sur-face hydrophobicity, which results in a low recovery rate of combustibles. More-over, due to the light weight and small particle size, the heterogeneous slime is entrained into the clean coal by non-inertial motion, further deteriorating the flotation process.
- Lines 48-52. The authors have indicated that SPP is selective to divalent metal ions such as Ca2+. After this, in lines 63-64, the authors claimed that “…it is expected that SPP has a good inhibiting effect on gangue, such as calcite and kaolinite”. According to Figure 1, X-ray diffraction analysis, the gangue contains quartz, kaolinite, and calcite. The SPP role as a depressant of calcite can be deduced; how acts the SPP over the kaolinite? Can it act selectively over the calcite?
Response:
The depression effect of the inorganic phosphate as a depressor was found mainly through the dispersion of gangue materials and the complexation of gangue materials, and these resulted in the improvement in the flotation selectivity [5]. Other than complexation with the gangue, the inorganic phosphate has been found to depress a number of gangue minerals in some studies [7-9]. We re-examined the statements of the third paragraph in the introduction section. This paragraph was poorly written, as it appeared to emphasize on the gangue complexation only, which did mislead the readers about the topic. Based on the comment, we revised the paragraph as follows.
Many studies have pointed out that inorganic phosphate can disperse silicate minerals, carbonate minerals and oxide minerals, which is believed that the addition of inorganic phosphates can increase the electrostatic repulsion and steric hindrance be-tween minerals. Theoretically, this dispersion can peel off and disperse the fine gangue adhered to the surface of coal particles, strengthen the recovery of coal particles, thus reduce the ash content of clean coal and enhance the flotation selectivity. Moreo-ver, inorganic phosphate is very easy to ionize in aqueous solutions and produce active anions to form colloidal complexions or chelate with divalent metal ions (such as Ca2+, Mg2+, Fe2+). It can reduce the surface hydrophobicity of these minerals and have a strong depressant effect, and further strengthen the selective recovery of the desired mineral. Some scholars have studied the inhibiting effect of inorganic phosphate in mineral processing. Wang et al. found that sodium tripolyphosphate (STSP) could strongly bind to Ca sites on the surface of calcite but weakly bind to Mg sites on the surface of magnesite. Therefore, STSP is considered as an effective flotation inhibitor for magnesite and calcite. Kang et al. studied a highly selective sodium hexameta-phosphate (SHMP) system, wherein SHMP was hydrolyzed into HPO42– and adsorbed on the positively charged surfaces of calcite and fluorite via electrostatic force or chela-tion. Based on the above literature, it is expected that SPP has a good inhibiting effect on gangue, such as calcite and kaolinite, in high-ash hard-to-separate coal. However, the application of SPP to improve slime flotation selectivity as a highly efficient dis-persant and depressant has not been extensively studied and has not been applied on a large scale.
In addition, we revised the title for the manuscript as well. In this study, the effect of SPP on flotation selectivity and the mechanism underpinning the effect were studied. Thus, the title has been changed to “Research on Mechanisms of Improving Flotation Selectivity of Coal Slime by Adding Sodium Polyphosphate”. The reason for this change is that we agreed that the original title “Selective Inhibition of Fine Slime in Coal Flotation with Sodium Polyphosphate” would also mislead readers. It is known to all that in coal flotation depressants would also reduce the combustible matter recovery to some extent. This study, however, indicates that with sodium polyphosphate only the recovery of ash materials was depressed, and the recovery of combustible materials was not influenced. This “selective” word is used for the ash materials and combustible matter, and that is why we call it “selective inhibition (depression). But there is possibility that other readers would take the “selective” as that the depression effect of sodium polyphosphate on a certain gangue mineral or some gangue minerals. To make it more precise, we changed the title of this manuscript.
- Table 1. It seems the columns of negative cumulative are unnecessary because the positive cumulative data is already included.
Response: The columns of negative cumulative includes yield and ash data. The sum of positive and negative cumulative yield is 100%, but the relationship between positive and negative cumulative ash content is not straightforward. To facilitate the readers’ reading and understanding, it was decided to keep the columns of negative cumulative. The positive cumulative data are available in many studies [1,2]. In addition, we revised the scientific terminology in the manuscript. In the revised manuscript, “Cumulative Oversize”, “Cumulative Undersize” and “Weight” are used instead.
- Figure 1. Some peaks in the diffraction pattern do not have an indexation, for instance: those between 30 and 40° or those for 2theta >50°. Are there other phases in gangue?
Response: Yes, we have added the indexation in Figure 1. Below are the details.
- Section 2.2.1 Flotation kinetics experiments. How many times was the experiment performed?
Response: The flotation kinetics test results were obtained by taking the average value through three parallel tests.
- Line 113. What is means ??∞? This is not included.
Response: The ??∞ is a typo. ?∞ is what we wanted to express. This has been corrected in the revised manuscript.
- The same units must be used in manuscript. For example, 1000 g/t (in line 100) and 1 kg/t (in line 128). The SPP content was expressed in g/t or mg/L. This is difficult for the analysis of results. Figure caption 1 includes that SPP used is 80 mg/L, which is the equivalent value in g/t or kg/t?
Response: The industry defines the dosage unit "g/t" as the mass of agent required per tonne of coal for flotation. The slurry concentration used in the test was 80 g/L, and for the reader's understanding, the unit of “g/t” is used throughout.
- Figure 1 must be located after being mentioned in the text.
Response: This has been addressed in the revised manuscript.
- Figure 2. Why done a model of the experimental data? Is this predictive? This is not discussed. It is notable in the graph that model lines show a deviation of experimental data even though the R2 is around 0.98; which is the error in the model?
Response: The Fuerstenau upgrading curves were used for characterization, comparison, and analysis of the whole coal flotation selectivity. These curves have proven to be very useful in the analysis of de-ashing and desulfurization of coal flotation results. Fuerstenau upgrading curves, which relate the recoveries of the two components in concentrate. A new parameter was defined as a separation selectivity index, the value k of which can be obtained by fitting the flotation kinetics using equation. It is generally accepted that the higher the R square value, the better the fitting. The deviation of experimental data and fitted data might indicate the errors in flotation data.
- Lines 201-202. “…By comparing the k values, the flotation selectivity increased and then decreased with the increase of SPP dosage”. In this case, how can be explained this behavior of the SPP on selectivity? For high SPP content (4000 g/t), the k value is lower even without SPP. How acts the SPP in gangue?
Response: The results indicate that the addition of SPP would change the flotation selectivity. The increase of SPP dosage to 1000 g/t resulted in the best flotation selectivity, but the further increase of SPP dosage deteriorated the flotation selectivity. As shown in many other studies, the excessive depressant act on the coal surface, which lead to a decrease in the recovery of combustible matter [3,5,8]. Thus, the flotation selectivity decreased as the SPP dosage kept increasing.
- Figure 3. SPP 80 mg/L at what amount of depressant corresponds? 1000g/t?
Response: The SPP dosage in laser particle size test has been mentioned in section 2.2.2. “According to the results of the flotation kinetics test, the optimum dosage of SPP was determined as 1000 g/t.” The“80 mg/L”was converted from the pulp concentration, the units have been unified in order to write a specification for the reader, which has been corrected in the manuscript.
- Some figures have low quality and must be improved for better visualization. For example Figure 4, the letters of the EDS points are not visible like the annotation of dimension. It is recommended that the contrast in the text be improved. Figure 9 it is not distinguished the text; the schema must also be improved.
Response: These have been addressed in the revised manuscript.
- Table 3 and text in lines 238-239. The authors describe that Al and Si content is diminished (except in spectrum D). These elements are associated with kaolinite and quartz; how is explained the role of SPP over these compounds? In all EDS results, the same elements are identified; what about the selectivity of SPP by calcite?
Response: One of the important roles of SPP is to increase electrostatic repulsion and spatial resistance between minerals to peel off and disperse fine gangue adhered to the surface of coal particles, to reduce the fine gangue coating (coverage) and to enhance the flotation selectivity.
To investigate the effect of SPP on fine gangue coverage, the energy spectrum analysis of coarse granular coal and its surficial fine particles was conducted. The composition analysis was used to determine whether the coated fine particles were gangue minerals. It should be noted that the main gangue in coal is kaolinite and quartz, and that explained why the content of aluminum and silicon was the focus.
Furthermore, due to the limited number of scans, it is impossible to accurately explain the effect of SPP on a particular mineral from a statistical point of view. However, it is certain that the gangue coating is significantly reduced by the action of SPP, and this dispersion can enhance the flotation selectivity. As for calcite, it has been speculated that its complexation of SPP, rather than physical adsorption, played the part.
14.Figure 4. The SEM images (c) and (d) correspond to samples after modification using SPP; this demonstrates the modification of surficial characteristics; nevertheless, how do the particles changes after the flotation process?
Response: The sample preparation for SEM tests is that, 1000 g/L SPP is added to the coal slurry with a concentration of 80 g/L. After 3 min, a certain amount of mineral slurry is evenly dripped on the glass slide and dried in a dust-free environment. The surface of coal particles before and after treatment was photographed by SEM. The direct purpose is to characterize the stripping of ultra-fine slime minerals off the coal surface by SPP. This reflects the subprocesses occurring in the process of coal flotation with SPP, which definitely allows the explanation of the decrease of gangue (ash materials) reporting to the concentrate.
- Figure 5. Which is the amount of the SPP used for this experimentation?
Response: We have added the conditions in section 2.2.4:“The interaction force measurements between solids were tested in pure water and 1% SPP solutions, respectively.”
- In section 3.3.1 Adsorption capacity of the collector. The text describes that 25.22 mg/L and 25.35 mg/L of collector are adsorbed in slime surface without and with the SPP use. The adsorption capacity must be reported in reference to solid mass (mg/g), not in concentration (solution).
Response: In literature, adsorption capacity is generally reported in terms of solid mass (mg/g). In this study, a slurry concentration of 80 g/L was used. Based on this, we revised the experimental adsorption capacity data.
- Figure 6. For this graph, which is the content of SPP used? Is the contact angle affected by the amount of SPP used?
Response: The effect of SPP on slime flotation showed that the SPP can enhance the dispersion and peel off fine slime minerals from the surface of coal particles. Besides, in this study, the dosage of SPP fixed at 1000 g/t was also selected to examine the effect of SPP on coal particle selectivity from the point of view of surface hydrophobicity. Yes, the coal contact angle increased with the addition of 1000 g/t, suggesting that the coal was more hydrophobic with the coated gangue removed from the coal surface by SPP. - Line 286. Figure 9 seems incorrect.
Response: Figure 9 is an illustrative diagram of the mechanism for SPP enhancing flotation selectivity. This figure highlights the fact that the addition of SPP enhanced the stripping and dispersion of the gangue from the coal particle surface. This resulted in a less coverage of gangue on the coal surface. With the action of collector, an increase in the hydrophobic sites on the coal surface was detected. The depression of the non-selective recovery of the gangue as well as the improvement of the adsorption of the collector on the coal surface are the SPP main roles, which lead to the improvement in the flotation selectivity in this study.
- Which are the contents of calcite, kaolinite, and quartz in gangue before and after the flotation process using the SPP? A characterization of concentrate and tailings after the flotation using SPP must be included to establish if the use of depressant is significant, as is claimed in the abstract and conclusions.
Response:
In this study, the effect of SPP on flotation selectivity and the mechanism underpinning the effect were focused. It has been found that the addition of SPP strengthened the stripping and dispersion of gangue minerals from the surface of coal particles. Literature also indicates the complexation of inorganic phosphates with Ca 2+, etc. Thus, it is concluded that the depressant effect of SPP on high-ash slime was a result of it reducing the slime coating, depressing calcite via complexation, as well as an improvement in combustible matter recovery by increasing the coal hydrophobicity with the aid of collector. Undoubtedly, that the contents of calcite, kaolinite, and quartz in gangue after the flotation process decreased using the SPP was obvious, as the ash content in the concentrate was found decreased. To be honest, the SPP interacting with different gangue minerals was out of scope of this paper, and this work is much more complicated to investigate in real ores where minerals are interlocked. This systematic research is now undergoing using the pure minerals, which will be reported in a new paper. But the relevant discussion has been added to the revised manuscript.
- What is the maximum content of gangue desirable in coal? Does an improvement in 2.39% of impurities justify the flotation process?
Response: In the field of slime flotation, slime ash is commonly used to characterize the content of gangue. The ash content of raw coal used in this study is about 24%, and the ash content of clean coal products required by coal preparation plants is usually less than 10%. By adding SPP, the ash content of clean coal is reduced by 2.39%. In coal flotation, even a reduction of 1% ash content is believed to be of great significance for the clean purification and utilization of high ash refractory coal.

Round 2
Reviewer 3 Report
The revised version provided by the authors has been improved. After the revision of changes made, I think that the authors have addressed all comments made by the reviewers. Then, I consider that the manuscript can be accepted in present form.